# Combination of Genomic Landsscape and 3D Culture Functional Assays Bridges Sarcoma Phenotype to Target and Immunotherapy

**DOI:** 10.3390/cells12172204

**Published:** 2023-09-04

**Authors:** Filomena de Nigris, Concetta Meo, Wulf Palinski

**Affiliations:** 1Department of Precision Medicine, School of Medicine, University of Campania “Luigi Vanvitelli”, 80138 Naples, Italy; concetta.meo@unicampania.it; 2Department of Medicine, University of California San Diego, La Jolla, CA 92037, USA; wpaliski@ucsd.edu

**Keywords:** sarcoma, 3D model, organoid target therapy, immunotherapy, clinical trial, genomic profiles, assay, organ on chip, explant

## Abstract

Genomic-based precision medicine has not only improved tumour therapy but has also shown its weaknesses. Genomic profiling and mutation analysis have identified alterations that play a major role in sarcoma pathogenesis and evolution. However, they have not been sufficient in predicting tumour vulnerability and advancing treatment. The relative rarity of sarcomas and the genetic heterogeneity between subtypes also stand in the way of gaining statistically significant results from clinical trials. Personalized three-dimensional tumour models that reflect the specific histologic subtype are emerging as functional assays to test anticancer drugs, complementing genomic screening. Here, we provide an overview of current target therapy for sarcomas and discuss functional assays based on 3D models that, by recapitulating the molecular pathways and tumour microenvironment, may predict patient response to treatments. This approach opens new avenues to improve precision medicine when genomic and pathway alterations are not sufficient to guide the choice of the most promising treatment. Furthermore, we discuss the aspects of the 3D culture assays that need to be improved, such as the standardisation of growth conditions and the definition of in vitro responses that can be used as a cut-off for clinical implementation.

## 1. Introduction

As emphasised by original contributions in this special issue, sarcomas are rare and heterogeneous neoplasms comprising more than 70 subtypes of bone and soft tissue sarcomas (STSs) [1]. Although they represent only 1% of malignant tumours [2], sarcomas account for 15% of childhood malignancies [2]. The standard treatment consists of surgery, with chemotherapy and radiotherapy as neoadjuvant or adjuvant regimes [2]. Given the rarity and histological variants and locations, clinical trials have not identified combination therapies effective against most sarcomas, and mortality remains at 40% for localised and 75% for metastatic tumour patients at five years [2].

In the past decades, the genomic and epigenomic characteristics of tumour subtypes have been a focus of research. The hope has been to better classify sarcomas on the basis of genomic rearrangements and to identify vulnerable sites to target treatments [3]. Drug rediscovery protocol (DRUP) indicated that rare tumours, such as sarcomas, benefit from genomics-guided treatment [4]. This approach has indeed delivered some successes, such as in lung tumours harbouring neurotrophic receptor tyrosine kinase (NTRK) fusion treated with its inhibitor [5]; in non-small-cell lung cancers (NSCLCs) treated with molecules targeting BRAFV600E mutations [6]; and in gastrointestinal stromal tumours (GISTs) treated with c-Kit inhibitors [7]. However, precision oncology approaches to cancer treatment have indicated that, in many patients, genomic profiling is not sufficient to match each patient to an effective therapy [4,8,9]. Most frequently, this is due to genomic sub-variants unresponsive to drugs, the development of drug resistance, or the attenuation of drug activity by the tumour microenvironment [10]. To address these limitations, strategies combining genome profiling with personalised ex vivo assays are needed. For this purpose, 3D models are preferable to traditional 2D monolayer cell cultures because they better reflect tumour heterogeneity and the interactions between the tumour cells and extracellular matrix with the tumour microenvironment. Similarly, patient-derived xenograft (PDX) models in which fresh patient tumour tissues are directly transplanted into immunocompromised mice are better than xenografts because they maintain the histological, epigenetic, and genetic characteristics. However, mouse models require at least 6–7 weeks for engraftment, which is a long time for experiments and has a high cost. While three-dimensional models generated from patient’s biopsy samples [11] are valuable assays, easy and quick to establish, and useful to (1) understand the genomic basis for each patient’s disease that fuels cancer, (2) match patients with the best treatment possible selected on the basis of the tumour’s genetic aberration, and (3) improve patient survival rates [12] (Figure 1). 

In contrast to novel clinical N-1 trials [13], based on single-patient treatment (see below), organoid assays provide fast responses and could, therefore, be routinely performed on a broader range of patients. Organoid assay platforms could comprise not only the data of responding and non-responding patients but also the data of controls, i.e., their results would be closer to real-world data (RWD). The larger number of organoid data from individual patients would also facilitate the definition of indices of ex vivo treatment effects. Indeed, despite the great potential of these models, the degree to which ex vivo assays reflect and predict clinical responses has yet to be established. In this review, we discuss how patient-derived ex vivo tumour models might be used as functional assays of precision oncology. We discuss important challenges that need to be addressed to enable their clinical implementation, provide an overview of current clinical studies using patient-derived 3D models to assess drug efficacy, and consider which parameters may enhance their potential ability to predict responses in patients.

## 2. Genomic Landscape as Biomarker for Diagnostic and Therapeutic Purposes

In order to understand the aims of the clinical studies discussed below, it is first necessary to provide a short overview of the genetic drivers of sarcoma pathogenesis identified to date. Bone and soft tissue tumours are generally classified on the basis of histology and genomic alterations promoting transformation into simple karyotypes characterised by gene fusion or complex karyotypes. All fusion genes are summarised in Appendix A. The pathogenesis of sarcomas with oncogenic fusion genes involves rearrangement between transcription factors, chromatin remodelling, epigenetic factors, or constitutive activation of serine/threonine or tyrosine kinases. Ewing’s sarcoma (ES) is a prototype characterised by *EWSR1:FLI1* fusion or *EWSR1* fusion with other partners [14]. In synovial sarcomas, the genomic translocation fuses *SS18* at 18q11 and *SSX1/SSX2* at Xp11 genes, resulting in an altered BAF complex [15,16]. The *YAP1:TFE3*-fused gene and *WWTR1(TAZ)-CAMTA1* gene fusions are characteristic of haemangioendothelioma, whereas the presence of *FUS:DDIT3* or *EWSR1:DDIT3* genes is pathognomonic in myxoid liposarcoma (MLPS), a malignant tumour, recapitulating lipogenesis [17,18,19]. The above driver oncogenes constitute the initial event and are sufficient to promote tumorigenesis. However, downstream of these rearrangements, many key pro-survival pathways are altered in sarcomas [20].

Sarcomas with complex karyotypes are a heterogeneous group that include angiosarcoma, myxofibrosarcoma, pleomorphic liposarcoma, osteosarcoma, undifferentiated sarcoma, leiomyosarcoma, and dedifferentiated liposarcoma. Large-scale genomic profiling studies have identified recurrent hotspot mutations, as shown in Appendix A. The deregulation of TP53 and RB1 genes is associated with the pathogenesis of sarcoma with complex karyotypes, together with variations in gene copy number variants (CNVs), mutational heterogeneity, and whole genome duplication, which favour metastasis, resistance to chemotherapy, and poor overall survival [21]. The loss of the oncosuppressor PTEN is found in 38.6% of sarcomas, most commonly leiomyosarcoma, osteosarcoma, chordoma, and epithelioid sarcoma [22]. Alterations in cell cycle genes due to the upregulation of their transcription factor, such as c-Myc, Forkhead Box F (FoxF1/FoxF2), and T-box transcription factor 3 (TBX3), are frequently found in leiomyosarcoma, osteosarcoma, chondrosarcoma, synovial sarcoma, and Ewing’s sarcoma (EwS) [23,24]. A second layer of uncontrolled activity is constituted by the aberrant activation of pro-survival and growth factor signalling pathways associated with genomic and epigenomic mutations. The most frequent receptors constitutively activated are platelet-derived growth factor (PDGF), epidermal growth factor (EGF), proto-oncogene tyrosine kinase receptor (c-KIT), insulin-like growth factor (IGF), and mesenchymal–epithelial transition (c-MET) pathways. All of these promote tumorigenesis by activating downstream Ras/Raf/MAPK and/or PI3K/AKT/mTOR pathways [25,26]. Proliferation and tumour progression in 50% of sarcoma cases are also associated with the aberrant constitutive activation of Yes1 associate protein (YAP) and Tafazzin (TAZ) transcription factors due to genomic alterations in the Hippo pathway [27]. Finally, another important pathway essential in tumour progression, and associated with patients’ poor prognosis, is angiogenesis, strongly activated in 25% of sarcomas through the upregulation of VEGFs and its receptors [2]. Taken together, the above findings indicate that many genomic alterations and several critical pathways have already been identified that promote the plasticity, aggressiveness, and chemoresistance of sarcomas and are targeted with existing drugs, as listed in Figure 2. 

## 3. Clinical Results of Targeted Therapy Based on Genomic Landscape

The standard of care for localised sarcomas consists of surgery associated with neoadjuvant (preoperative) or adjuvant (postoperative) chemotherapy or radiation [2,28]. However, the percentage of recurrence is around 20% of cases, and up to 50% develop metastases [28]. For metastatic sarcomas after chemotherapy, the median overall survival is 12 months, and only 10% of patients have a 5-year survival rate [28,29]. Targeted therapy is an important approach to overcome the current therapeutic limitations, because it is directed against the genomic rearrangement or mutations that period influence cancer growth. Recent studies established that genomic profiling of patient-derived primary sarcoma cell cultures is the best tool to identify biomarkers for the stratification of patients and mutations or rearrangement suitable to target treatment [30,31,32]. Additionally, the authors of these studies investigated intra- and inter-genomic variability of two sarcoma histotypes myxoid fibrosarcoma (MFS) and undifferentiated polymorphic sarcoma and identified the signatures that could be partially responsible for different chemo-susceptibilities or chemoresistances [31,32].

In this section, we review clinical trials using targeted therapy selected on the basis of actionable mutations identified by genomic profiling of patients (Appendix A).

### 3.1. Cell Cycle and MDM2 Inhibitors

Amplification of cyclin-dependent kinases (CDKs) is typical in sarcomas and present in over 90% of well-differentiated/dedifferentiated liposarcomas (WD/DDLPSs). Many efforts have been made to develop CDK-specific inhibitors with low toxic effects. Currently, the most effective molecules include the CDK4/6 inhibitors Palbociclib, Ribociclib, and Abemaciclib. Palbociclib, in particular, has demonstrated antitumor activity in patients with advanced or metastatic WD/DDLPS, improving survival (NCT01209598) [33]. Palbociclib showed efficacy in osteosarcomas with a loss of pRB function and/or the amplification of CDK4/6, whereas most other histologies were resistant [34]. To overcome these challenges, ongoing clinical trials are combining chemotherapy with CDK inhibitors (NCT04129151, NCT03709680, NCT02897375, NCT02784795, NCT03009201, NCT03114527, and NCT02343172). Another gene often duplicated and overexpressed in sarcomas is *Mouse double minute 2 (MDM2)*, which mediates p53 ubiquitination and its degradation [35]. Several inhibitors of MDM2 were developed, such as nutlin-3 and RG7112. When used alone or in combination with CDK4, they significantly reduced tumour growth in liposarcomas with amplified MDM2, as reported in a Phase I clinical trial [36]. The most promising small molecular compounds inhibiting the MDM2–p53 interactions are nutlins. Nutlins were tested in osteosarcomas in combination with gemcitabine, doxorubicin, CDK inhibitors (roscovitine), and aurora kinase inhibitors [37]. However, recent data indicate the increasing development of drug resistance [38]. Many small compounds were synthesised to target p53 and its mutant isoforms. The most widely investigated of these molecules are PRIMA-1(2,2-bis(hydroxymethyl)-1-azabicyclo(2,2,2) octan-3-one) (also known as APR-017) and PRIMA-1MET (also known as APR-246), which target wild-type p53 and downstream targets, including p21, Noxa, Puma, GAD45, and MDM2 [39]. Recent data have indicated the acceptable safety profile and encouraging activity of APR-246 in combination with azacytidine, supporting further frontline evaluation of this combination. Some molecules have been selected for their capability to target fusion transcription factors in sarcomas. One example is Trabectedin, which interferes with the ability of FUS-CHOP protein to bind its target promoters [40,41]. Trabectedin was clinically effective in treating leiomyosarcomas and liposarcomas and is currently in clinical trials for other sarcoma subtypes (NCT02367924, NCT02275286, NCT04076579, NCT01303094, and NCT04067115) (Appendix A).

### 3.2. Signal Transduction Inhibitors

The PI3K/AKT pathway and downstream signalling are activated constitutively in the majority of localised osteosarcomas and are involved in their progression [42]. BKM120 (Buparlisib) is a novel, less toxic PI3K inhibitor showing antiproliferative and apoptotic effects in osteosarcoma, Ewing’s sarcoma, and rhabdomyosarcoma cells [43]. Preliminary clinical evaluation has indicated that it is active in several malignancies, including sarcomas [43,44]. In preclinical osteosarcoma studies, BKM120 was shown to block cell proliferation and determine cell death by suppressing the PI3K/AKT and MAPK/ERK pathways [43]. Non-randomised Phase II clinical trials have evaluated mTOR inhibitors that act by interrupting the PI3K/Akt/mTOR pathway. Everolimus, a selective mTOR inhibitor, was used to treat unresectable high-grade osteosarcomas after standard treatment with Sorafenib [45]. The results indicated that 45% of osteosarcoma patients were progression-free at 6 months. Similarly, in another multicentre Phase II trial, Everolimus reduced tumour progression in 29% of patients affected by metastatic, recurrent bone, and soft-tissue sarcomas [46]. Tyrosine kinase inhibitors (TKIs) represent a highly successful targeted therapy for sarcomas. In the PALETTE Phase III trial, Pazopanib targeting VEGFR and both c-KIT and PDGFR receptors improved patient progression-free survival by 3 months and was beneficial in the treatment of resistant metastatic sarcomas [47]. In contrast, Pazopanib showed poor efficacy in the preoperative treatment of STS patients in another Phase II clinical trial (GISG-04/NOPASS) [48].

In desmoid tumours, prolonged progression-free survival was reported following Sorafenib treatment in combination with another multi-TKI [41,49], and favourable disease-control rates were observed in leiomyosarcoma, synovial sarcoma, and malignant peripheral nerve sheath tumour (MPNST) patients [50]. In contrast, moderate Sorafenib activity was reported when it was used as second-line therapy in metastatic sarcomas [51], or in combination with chemotherapy, due to toxic side effects in advanced STS. Confirmed toxicity and other severe adverse effects were observed with Pazopanib, Sorafenib, and Sunitinib treatment in several clinical trials [52], limiting their utility. Fewer side effects were reported for ZD6474, a small TK inhibitor that targets VEGFR-2 with promising results in vitro and in vivo mice models, but no trials have been reported [53]. For the treatment of gastrointestinal stromal tumours (GISTs), the FDA approved the targeted drug Imatinib, a TKI that inhibits both c-KIT and PDGFR. It showed great efficacy and is currently used as the first-line treatment for GISTs [54] and dermatofibrosarcoma [55]. Other small molecule inhibitors of c-KIT, such as Selinexor, Regorafenib, and Cabozantinib, have shown mixed results in different soft tissue sarcomas [56,57,58] (Appendix A). Neurotrophic receptor tyrosine kinase (NTRK) inhibitor is another success of targeted sarcoma therapies. Currently, two small molecule inhibitors of TRK are approved for severe cases of sarcoma in which NTRK gene fusion is present: Larotrectinib and Entrectinib [59]. The results of these studies suggest that the genomic profiles of advanced solid tumours are useful for identifying mutations but often do not match with selective treatment. Therefore, the addition of timely preclinical assays that help to better define patients’ sarcoma subtypes and identify those that respond to specific treatments seems necessary to truly implement precision oncology in the clinical routine (Appendix A).

### 3.3. Current Immunotherapy and Clinical Results

Tumour immunotherapy can be divided according to mechanism into targeted therapies based on antibodies, checkpoint inhibitors (ICIs), cytokine therapy, adoptive T-cell transfer (ACT), and tumour vaccination. The use of monoclonal antibodies targeting receptors as monotherapy or in combination with chemotherapy for sarcomas has demonstrated limited activity or did not improve overall survival. For example, the combination of Olaratumab, a monoclonal antibody against platelet-derived growth factor receptor alpha, with doxorubicin provided no detectable clinical benefit in terms of PFS and overall survival in osteosarcoma compared to conventional chemotherapy [60]. Monoclonal antibodies against IGFR-1, such as Cixutumumab, provided modest clinical benefits as the second line of treatment for Ewing’s sarcoma, advanced metastatic liposarcoma, osteosarcoma, rhabdomyosarcoma, and synovial sarcoma patients in Phase II clinical trials. Additionally, patients who initially responded to immunotherapy developed drug resistance and suffered disease recurrence [61].

Potentially better outcomes compared to conventional chemotherapy were reported in some sarcoma histotypes with immune checkpoint inhibitors (ICIs). Sarcomas with a high frequency of chromosome copy number alterations, such as UPS and MFS, may be capable of eliciting an immune response and respond better to ICIs or other immunotherapies [62]. However, most histology types do not benefit from treatment, underlining the necessity to identify biomarkers predictive of ICI response. Petitprez et al. developed a new classification of STS based on the composition of the TME, which varies according to histology and is generally immunosuppressive [63]. In this study, tumours were assigned to one of five sarcoma immune classes (SICs) based on the microenvironment cell population (MCP) determined by the FACS counter. The SIC system predicted responses to PD-1 blockade in a multicentre phase II clinical trial in STS (NCT02301039) [64].

One mechanism contributing to the immunosuppressive effect of the TME is the downregulation of the HLA immunophenotype inhibiting CD8^+^ cell activation following PD-1/PD-L1 treatment [65]. However, clinical studies reported responses to ICIs in only a few sarcoma entities [66]. Patients affected by undifferentiated pleomorphic sarcoma and alveolar soft part sarcoma showed the highest overall response and non-progression rate, whereas the lowest overall response was observed in leiomyosarcoma patients [67]. In contrast, monotherapy with a CTLA-4 inhibitor or in combination with Ipilimumab was ineffective in patients with synovial sarcoma [68]. This suggests that the success of anti-PD1/PD-L1 treatment is largely dependent on specific sarcoma subtypes [67]. The combination strategy is of particular interest when it targets the immunosuppressive tumour microenvironment and enhances antitumor immune responses. Recently, the IMMUNOSARC study, a Phase II clinical trial, investigated the efficacy of TKI inhibitors, such as Sunitinib or the PD-1 inhibitor Nivolumab in the treatment of advanced STS. The results indicated that each drug improved overall and progression-free survival at 6 months in 77% and 50% of cases, respectively (NCT03277924) [69]. Current studies are using monoclonal antibodies against PD-L1 and CTLA-4 to treat rare sarcomas, such as DDLPS and pleiomorphic liposarcoma (NCT02500797 and NCT03114527) or the effect of the PD-L1 inhibitor Durvalumab in combination with either Pazopanib (a TKI inhibitor) or Tremelimumab (a CTLA-4 inhibitor) in advanced STS (NCT03798106 and NCT02815995).

Another approach to immunotherapy is chimeric antigen receptor T (CAR-T) adaptive cell therapy. This involves isolating the patient’s own T-cells, modifying them to express a CAR that recognises a specific tumour antigen, and then reinjecting them into the patient [70]. The recognition of the tumour cells by the CAR activates T-cell proliferation, which may result in tumour lysis. In addition to a number of in vitro studies indicating the success of this approach, one completed and four ongoing clinical trials looked at sarcoma patients. One of the ongoing Phase I/II trials uses epidermal growth factor receptor 2 (HER2) CAR-T cells to treat advanced-stage ewing’s sarcoma (ES), osteosarcoma (OS), rhabdomyosarcoma (RMS), desmoplastic small round cell tumour (DSRC), and peripheral neuroectodermal tumour positive to the HER2 receptor after chemotherapy (NCT00902044). The initial data showed that this approach improved the median survival of patients from 5.1 to 29.1 months without inducing toxicity [71]. Additionally, a Phase I clinical trial that utilises CAR-T cell therapy to target EGFR- and CD19-positive tumours to treat children and young adults with recurrent/refractory solid tumours is ongoing (NCT03618381). Promising results were also reported in a study in which T-cells from standard myxoid liposarcomas were genetically engineered to recognise NY-ESO1, an antigen expressed in 80–90% of cases and in 70–80% of synovial sarcomas (NCT02992743) [72]. NY-ESO1 T-cells recently gained FDA approval for patients with advanced synovial sarcoma [73]. In summary, preclinical studies demonstrated that CAR-T cell therapy and targeting sarcoma-associated antigens are effective, and one concluded plus several ongoing clinical trials are further evaluating their therapeutic effects (Appendix A).

## 4. Clinical Trials Based on Genomic Landscape and N-1 Trials

Trials on rare cancers are usually designed and conducted in small genetically or biomarker-defined subsets of patients. However, the restrictive criteria and the limited analyses answer only specific questions but may not yield real-world data (RWD) and real-world evidence (RWE). RWD data include information on the standard care of patients obtained from prospective and/or retrospective observational studies [74], and RWE should reflect the actual results obtained for drug treatments. Small trials have also highlighted the problem of appropriate controls, which may delay patient access to lifesaving therapies and may not be ethical. One way proposed to overcome this is a synthetic control arm external to the study [75]. The external control of this study provided evidence of the natural history of the disease, which helped to establish that the observed responses and improvements in symptoms constitute a genuine treatment effect [76]. Nevertheless, external control arms remain a concept and are generally lacking in rare tumour trials.

The lessons learned from these studies have influenced the design of different classes of trials. Currently, umbrella, basket, platform, and master observational trials (MOTs), as well as platform trials, put the patient’s genomic landscape at the centre of targeted therapy and attempt to speed up treatment decisions [25]. Important insights are emerging from such trials: (1) single-agent therapy is not sufficient to determine tumour regression; (2) actionable driver mutations respond differently to treatment, depending on the tumour microenvironment; and (3) tumours adopt many mechanisms of resistance. The complexity of molecular pathways activated downstream of the genetic alteration in the tumour affects individual responses. One possible answer could be a combination of anticancer molecules and genomic-targeted agents. Because the number of potential combinations is very large, the best strategy is to select the most promising drugs on the basis of strong preclinical data. Models to obtain such preclinical data include cell line xenografts in animals, patient-derived xenografts (PDXs), and patient-derived three-dimensional models, with all testing drugs at clinically used doses. Based on these prerequisites, Combo-MATCH, a large precision medicine trial, was launched in 2019. It combined treatment with drugs targeting actionable mutations selected on the basis of genomic profiles of individual patients, with other molecules selected on the basis of preclinical evidence. The aim is to determine whether combination therapies are more efficient than single-target agents and whether preclinical models predict clinical outcomes. More specifically, this innovative trial combines genomic therapy and precision Phase I safety data generated by timely ex vivo patient assays or by patient xenograft models to treat individual patients [77].

Another tool to study personalised treatment is NCI-MATCH-N-of-1 trials. An N-of-1 trial is a single-patient clinical trial to evaluate the efficacy and/or adverse events of one or several interventions chosen on the basis of individual molecular and clinical data [78]. So far, only a few N-1 studies on gene therapy for rare diseases have been reported [79,80]. However, these raised important questions, such as what level of evidence is needed before exposing a human to a targeted drug, and what evidence would be sufficient to lead to generalised treatment [81]. The need for patient avatars capable of providing timely information on drug response is, thus, becoming ever clearer (Figure 3).

## 5. Individualised Patient Models

### 5.1. Spheroid Models

Spheroids are extensively used in cancer research. They can be generated from tumour cell lines or by culturing primary tumour cells, allowing their aggregation into small 3D clusters using different methods [11]. Many protocols have been established to increase their cellular complexity, e.g., by adding clusters of immune and stromal cells. These multicellular spheroids aim to reproduce the tissue characteristics of the original tumour for some time (days to weeks) [82,83,84]. However, spheroids have a limited capacity to proliferate, do not accurately reflect the cellular composition of the tumour and its tissue characteristics, and the amount of starting material directly determines the number of spheroids that can be generated. These limit the clinical use of spheroid models.

### 5.2. Spheroid Assays to Test Chemo- and Targeted Therapy in Preclinical and Clinical Trials

Various preclinical studies have utilised sarcoma spheroids of different histologies as models to study responses to chemotherapies and target therapy in osteosarcoma [85], chondrosarcoma [86], rhabdomyosarcoma [87,88], and Ewing’s sarcoma [89].

Most frequently, spheroids were used as platforms to screen a panel of cytotoxic agents approved for other tumour types, and the threshold of spheroid cell death was used as a parameter of drug efficacy. For example, in one study of Ewing’s sarcoma (EwS), spheroids were used to mimic the temporal sequence of chemotherapy administration in patients, using a microfluidics droplet system. Spheroids generated inside droplets showed similar sizes and low intratumoural differences, and growth and viability could be measured by using fluorescence markers of live and dead cells. Data from an ES model showed that sequential combination treatment with etoposide 24 h before cisplatin resulted in a synergistic tumour cell death effect. The authors, thus, identified the droplet-based microfluidics approach as a valid tool for the evaluation of drug combinations [90]. In this study, 55% of the spheroids classified as responders predicted the same response in patients, whereas the rest showed stable or progressive disease. Myxoid liposarcoma (MLPS) is a lipogenic sarcoma, characterised by myxoid appearance and the presence of the *FUS-DDIT3* fusion gene. MLPS frequently recurs and has a poor prognosis after standard treatments, such as surgery, and is, therefore, in dire need of novel therapeutic approaches. In a preclinical study, spheroids established from two MLPS patient cell lines (NCC-MLPS2-C1 and NCC-MLPS3-C1) were successfully used for high-throughput drug screening of cytotoxic drugs. Of the 213 tested anti-cancer agents, Romidepsin was selected as the most efficient to suppress cell proliferation, even though it had the lowest IC50. These results indicated that spheroids are a useful tool for basic and preclinical studies of MLPS and probably for other rare tumours, too [91].

Radiation is an important treatment for sarcomas, particularly those resistant to chemotherapy. The efficacy of gadolinium oxide nanoparticles (GdoNPs) in combination with single (4 Gy) or fractionated (4 × 1 Gy) irradiation was evaluated in chondrosarcoma spheroids and in vivo in a mouse model. Gadolinium oxide treatment decreased spheroid growth and murine tumours. Based on these results, a multicentre, randomised Phase 2 trial is currently evaluating the combination of GdoNP with radiotherapy for the treatment of patients with inoperable musculoskeletal tumours [92]. The utility of transcriptomics analysis to identify personalised therapeutic targets was also demonstrated in Ewing’s sarcoma. A spheroid model was used as a functional assay to test a target therapy selected by epigenomic and genomic screening. Spheroids were treated with a bi-functional single molecule inhibiting both PARP and HDAC [82,93,94]. Although statistical significance was obtained only for a few patients from which cell lines were obtained, the results illustrated the potential of spheroids to reflect individual patient responses to therapy.

Spheroids can contain multiple cell types present in the tumour microenvironment (TME) and may, therefore, be useful for evaluating patient responses to immunotherapy. Spheroids generated from chordoma with a diameter of 40–100 μm contained immune cells, such as T lymphocytes (CD3^+^, CD4^+^, and CD8^+^), and myeloid cells (CD11b^+^ and/or CD11c^+^) [83]. They also responded to anti-PD-1 antibodies, providing evidence that the antibody was able to activate T-cells in culture, which led to subsequent tumour cell death [83,95]. However, it remains to be established whether responses seen in spheroids are correlated with those seen in patients. Moreover, major challenges for spheroid models remain. Given their short survival time, the assay results reflect only short-term effects of drug intervention, and the number of spheroids is limited by the amount of tumour tissue retrieved (Table 1).

### 5.3. Organoid Models

Patient-derived organoid (PDO) protocols have been established for many primary and metastatic sarcomas [96]. Several techniques have been developed to culture biopsy tissues [12]. Generally, patient-derived tumour tissues are dissociated into single cells, including stromal and immune cells, and then embedded in different materials, most commonly hydrogels. Mixed cells are then grown in culture media containing growth factors [12]. In this environment, all cell components are able to spontaneously organise into 3D structures as the original tumour, enabling cellular expansion and longer culture (weeks) compared to spheroids [97]. PDOs recapitulate and preserve the genetic diversity and heterogeneity of the originating tissues and can be stored [98,99]. Multiple “living biobank” initiatives have been started to create collections of large numbers of PDOs. Such systems may aid drug development and testing in normal or cancer tissues.

### 5.4. Organoid Assays to Test Targeted Drugs and Chemotherapy in Preclinical and Clinical Trials

Organoids are widely explored as functional assays to guide the selection of target treatments and chemotherapies. In most studies, organoids are prepared from surgery specimens in order to have sufficient material, and drug screening is performed a few days before starting patient treatment. Several reports have established correlations between responses in PDOs and responses in patients, including disease trajectory, and have helped to characterise the landscape of drug resistance [100,101,102,103]. A larger collection of tissues (194 specimens from 126 sarcoma patients, spanning 24 distinct subtypes including metastases) was recently evaluated to establish whether they could be used to predict treatment sensitivity. Organoids generated from these tissues showed patient-specific growth characteristics and subtype-specific histopathology. The genetic features of the tumours identified actionable mutations and indicated several biological pathways implicated in the response to treatment. Of all organoids tested, 59% were sensitive to some of the target drugs tested, and screening provided additional information on drug resistance useful for avoiding ineffective treatments [100]. However, the correlation between ex vivo data and clinical treatment responses was not determined, and we do not know the predictive value of the results in the PDO subgroup treated with FDA-approved drugs. The study also highlighted the need to predefine the cut-off values for drug responses in PDOs and to have results in time for treatment decisions, as indicated in Figure 3.

Rhabdomyosarcomas (RMSs) are mesenchymal-derived tumours and the most common childhood soft tissue sarcomas. Despite intense treatment, the prognosis for high-risk patients is poor. The discovery of new therapies would benefit from additional preclinical models. Recently, 19 paediatric organoids from all major RMS subtypes have been generated. Molecular, genetic, and histological characterisation showed that the models closely resembled the original tumours and were genetically stable over extended culture periods of up to 6 months. Tumours of mesenchymal origin can, therefore, be used to generate organoid models relevant to a variety of preclinical and clinical research questions [104]. Novel trials are underway, such as OPTIC (NL6166804117), which is attempting to systematically evaluate correlations between PDOs and clinical responses with standard of care (SOC) therapies in patients with metastatic disease (Table 1).

Recently, a multi-cohort proof-of-concept study (NCT04986748) involving sarcoma and melanoma patients has started. This trial will use 3D patient models as functional screening of a 14-drug panel, and the in vitro results will indicate patient treatment. Another trial of interest is Sarco PDX (NCT02910895), which will use organoids generated from preoperative biopsies and patient-derived xenografts to personalise the screening of standard chemotherapy in combination with tyrosine kinase inhibitors in advanced and metastatic sarcoma [105]. However, no data have yet been reported. In a closed osteosarcoma trial (NCT03358628), patient tumour tissues were implanted in nude mice to evaluate the efficacy of chemo- and targeted therapy selected on the basis of genomic and epigenetic modifications. The data reported that patient-derived xenografts (PDXs) effectively predict patient response in 17 regimens of treatment, suggesting that PDX is a satisfactory model with a high engraftment rate and accuracy in its prediction of drug efficacy. However, PDX models require a long time to obtain drug screening results. Finally, the PIONEER study will establish a tissue biobank of high-quality biological specimens and will associate clinical data with large-scale patient-derived organoid drug screening to support drug discovery, diagnostic assay development, oncology biomarker discovery, and other purposes. Additional studies are listed in Table 1. In summary, extensive ex vivo studies and many ongoing studies will provide valuable clinical information, specimens, and organoid assay data to improve sarcoma treatments.

### 5.5. Organoid-Based Assays to Test Immunotherapy in Clinical Trials

Although chemotherapies remain the primary treatment for most sarcomas, immune checkpoint inhibitors (ICIs) have been increasingly used. At present, ICI treatment is recommended when there is evidence of increased PD-1 or PD-L1 expression, deficiency in mismatch repair proteins, or increased tumour mutational burden (TMB). Despite these indications, not enough evidence supports the use of ICIs as first-line monotherapy for advanced sarcomas, because ICIs often fail to induce long-lasting efficient cytotoxic responses. In contrast, the combination of chemotherapy with immunotherapy has shown promising results in unresectable or metastatic angiosarcoma and leiomyosarcoma subtypes [106]. Ongoing trials are testing different combinations of cytotoxic drugs with ICIs (NCT 03899805, NCT 03123276, and NCT 04332874), nivolumab (NCT 04535713 and NCT 03590210), or durvalumab (NCT 03802071) [107].

The ability to predict patient-specific responses to immunotherapy with functional assays would benefit them both in terms of timely interventions and cost effectiveness. For this purpose, several organoid-based co-culture models designed to reconstitute or preserve the immune system have been developed. One of the few studies on chordoma organoids demonstrated that they may recapitulate immunosuppressive microenvironments. In particular, chordoma organoids showed the typical immune exclusion phenotype and macrophage M2 polarisation observed in patients. Studies of the culture media from organoids detected the secretion of CCL5 chemokine [108] and its knockdown, and treatment with MVC (a CCL5/CCR5 inhibitor) both significantly inhibited the progression of malignant chordoma and M2 macrophage polarisation. This suggests that the CCL5-CCR5 axis is a potential therapeutic target and further supports the notion that organoids are a valid model for studying the microenvironment of rare tumours [108]. Correlations between PDO-specific T-cell reactivity and clinical responses to immunotherapy will be evaluated in an ongoing clinical study involving paediatric patients (NCT058907813). One of the first studies including hundreds of immune organoids of different histologies was provided by Neal and colleagues. These authors adopted a novel technological approach based on the air–liquid interface of the collagen matrix to produce organoids representative of the immune compartment with syngeneic T-cells [109]. In this system, the TME was found to be preserved, including the presence of CD8^+^ and CD4^+^ cells, tumour-infiltrating lymphocytes (TILs), and stromal cells. The authors observed infiltrating CD3^+^ T-cells expressing PD-L1 and the reactivation of TILs following treatment with ICIs. Unfortunately, no comparisons were made between these PDOs and patients receiving the same treatments. Future prospective studies will be required to establish correlations. Of note, when organoids did not contain sufficient intertumoral immune cells, it was necessary to add peripheral blood mononuclear cells to the co-culture in order to detect immune responses. These studies constitute an important first step towards the implementation of precision immuno-oncology. Additional clinical trials are listed in Table 1.

### 5.6. Explants on Chip Sensor (Organ on Chip) Assays in Preclinical and Clinical Trials

Explant cultures consist of small tumour biopsy fragments (typically 1–2 mm^2^) cultured ex vivo [110]. They aim to maintain the native tissue architecture and microenvironment and to preserve the immune cell composition, such as stromal cells, lymphoid cells and, at least in part, myeloid cells [111]. However, the viability of the cellular component is limited to a brief period (2–5 days), and drug screenings have to be performed within this timeframe. Furthermore, the number of explants from individual patients is limited by tissue availability, which often prohibits large-scale drug screening. The most innovative system to grow the explants is chip devices. Tissues or 3D aggregates of multiple cell lines can be grown on transparent flexible polymers the size of a computer chip, called “organ chips”. These bionic models incorporate perfusion devices to create precise control of fluid flow, which allows an accurate distribution of drugs over time and long periods of culture (weeks or months, depending on the cell type) [112]. Some organ chips also include an endothelial channel mimicking the luminal surface of blood vessels, which can be perfused with immune or cancer cells, or even whole blood [113]. Early applications of these chips have been reviewed elsewhere [112]. Using a custom-built polystyrene device, researchers are now able to reproduce many aspects of the tumour microenvironment, cell proliferation, glucose uptake, and oxygen consumption, as well as the presence of necrotic regions, and use them for drug tests [114].

To date, no clinical trial results on sarcoma explants have been reported; however, a particularly important study is that of Majumder and colleagues, who compared responses to combination treatments in organs on chips to clinical trial results using the CANSCRIPT platform. In this study, 109 biopsies from head and neck squamous cell carcinoma (HNSCC) and colorectal carcinoma (CRC) were co-cultured with autologous plasma and peripheral blood mononuclear cells [115]. Docetaxel, cisplatin, and 5-FU were used to treat patients and biopsies for HNSCC or Cetuximab plus FOLFIRI (folinic acid plus 5-FU and Irinotecan) for CRC. Functional ex vivo results, such as viability and cell death after treatment, were determined, as well as clinical responses. All findings were then used to train a novel machine learning algorithm to distinguish responders to therapy from non-responders. The powerful CANSCRIPT platform was able to identify patient responders to therapy with high specificity. These results strongly support the notion that with proper study design (a large enough study population), the chip platform is a powerful tool for classifying patients. The extension of the clinical trial arm of the study may also reveal the degree to which positive ex vivo treatment effects predict beneficial effects in patients.

Another area in which important progress has been achieved is immunotherapy. Tumour explants that preserve both inter- and intra-tumoural immune clusters are of particular interest for immunotherapy, not only as a prognostic tool but also to select predictive biomarkers of responses to anti-PD-1 antibodies [116]. To date, no studies on sarcomas have been reported, but relevant insights are provided by data for melanoma, NSCLC, breast and ovarian cancers, and renal cell carcinomas exposed to anti-PD-1 antibodies [117]. A comparison of the ex vivo responses with real-world clinical data showed a strong correlation in 50% of cases. Furthermore, the platform used for the functional investigation of responses to anti-PD-1 antibodies led to the identification of several TME “immunotypes” associated with different responses to anti-PD-1 antibodies [117]. A similar study using tumour explants 30–450 μm in diameter was also performed with NSCLC. It demonstrated that following anti-PD-L1 treatment, explant tissues and patients released interferon. However, no other comparisons were made between explants and patients [118]. The short time of explant viability unfortunately impeded the analysis of other outcomes, such as T-cell expansion or T-cell killing. Data from these lines of research highlight the ability of explants to reproduce early patient responses to immune checkpoint inhibitors and suggest that the models are a valid tool for precision immuno-oncology [119]. Additional clinical trials are reported in Table 1. The predictive value of tumour explants for immunotherapy will, nevertheless, depend on the continuous innovation of assays. Several novel approaches are currently being explored that enable both automation and high-throughput strategies combining microfluidics, bioengineering, and nanotechnology for the development of lab-on-a-chip devices as promising platforms for drug screening [120,121].

## 6. Other Potential Evolutions

It is also conceivable that future developments to find novel tumour treatments could use 3D models generated not from individual patients but from genetically modified normal or tumour cell lines. In this case, the genetic modification is identified in a range of different tumours by the existing data from large-scale genomic or mutational studies. Using such models may provide clues about the relevance of the genetic alteration for tumour growth and indicate their susceptibility, in principle, to a broad range of novel compounds without quantitative or ethical constraints.

## 7. Prospective

Despite the identification of genomic profiles and aberrant mutations associated with sarcoma pathogenesis, the success rate of interventions targeting some of them has been very low. The highly variable histology of their subtypes and their rarity pose many challenges discussed in the present review. The development and implementation of different three-dimensional tumour models as functional assays to evaluate the drug vulnerability of individual patients and guide their treatment have not yet received clinical validation. Many limitations impede their use as valid tools. One of the issues is the assay time. Three-dimensional models need to be established from patient biopsy early in the disease trajectory, i.e., before any tumour-directed clinical treatment begins, to avoid alterations in tumour characteristics. Furthermore, to improve the statistical significance and predictive accuracy of the ex vivo functional assays, a great number of PDO cultures need to be generated from individual biopsies, preferentially surgical biopsies. The type of 3D model should be decided on the basis of the clinical question. Neither spheroids nor explants can survive multiple cycles of expansion, and drug tests, therefore, must be performed a short time after biopsy sampling. On the other hand, organoids grow better than spheroids and are generally preferred as a platform to model the tumour microenvironment and for personalised immunotherapy or genomic target therapy.

For all these models, standardised quality criteria must be established, the conditions of cultures predefined, and algorithms created to determine the cut-off of a treatment effect in vitro, which should be supported by evidence. Furthermore, several reports have indicated a high degree of both inter-patient and intra-patient heterogeneity among different organoid platforms [122,123]. This complicates the translation of assay results to patients. Creating standardised assays while preserving as much as possible of the original tumour properties is, therefore, important. The basement membrane extracts supporting 3D cultures, such as Matrigel or Geltrex, could also impact the assay. Matrigel is a sarcoma-derived extracellular matrix and contains growth factors and cytokines that promote growth. Synthetic polymers, such as hydrogels or other materials with well-defined structures and inert properties, may overcome alterations in growth [124]. Collagen-based scaffolds appear to better mimic the microenvironment of some rare sarcomas [125,126]. The most difficult problem will probably be to reach an agreement on what cut-off of drug response in each of the assay models should determine patient treatment. Despite these challenges, the concept of functional assays holds great benefits to improve precision medicine, and technological advances might help to overcome the challenges.

## 8. Conclusions

The results of completed and ongoing trials highlight the challenges associated with targeting genetic mutations. Optimizing treatment options on the basis of personalised preclinical assays will offer better treatment to patients and unlock the full potential of precision oncology.

## Figures and Tables

**Figure 1 cells-12-02204-f001:**
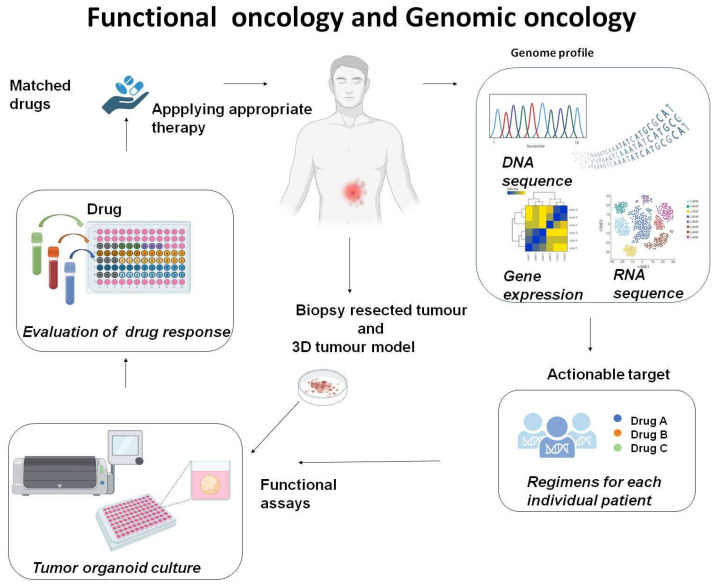
Bridging genotype to phenotype. Genomic profiles can help to identify actionable mutations and vulnerable pathways targetable with multiple drugs. Patient-derived tumour models can be used for drug screening assays providing functional data with the potential to guide the selection of specific therapies, thus improving likely patient outcomes and avoiding overtreatment and toxicities.

**Figure 2 cells-12-02204-f002:**
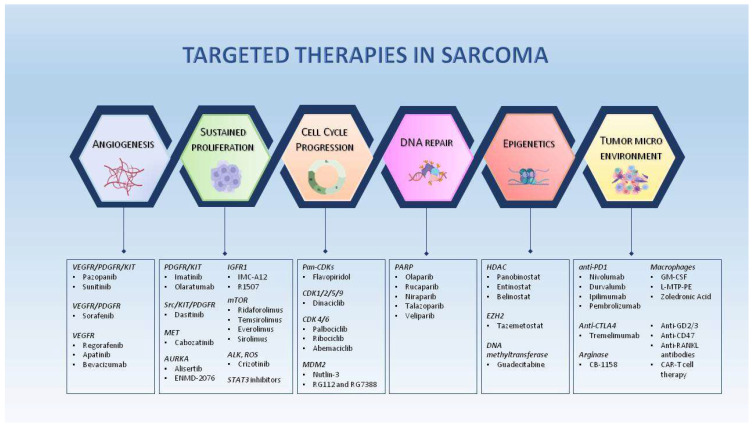
Targeted therapies for sarcomas. The illustration shows molecular pathways promoting sarcoma oncogenesis, including cell cycle progression, DNA repair, epigenetics, tumour microenvironment, and angiogenesis. Boxed in blue is a selection of approved drugs and their respective targets.

**Figure 3 cells-12-02204-f003:**
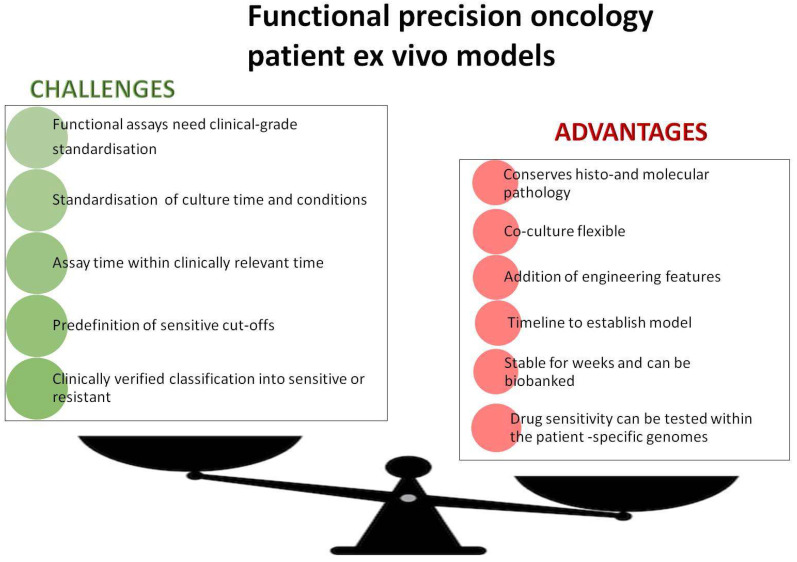
Current limitations and advantages of functional assays based on ex vivo three-dimensional tumour models.

**Table 1 cells-12-02204-t001:** Clinical trials including organoids for functional precision medicine (source: www.clinicaltrials.gov, accessed on 23 July 2023).

Clinical Trials Including Organoid Platforms					
NCT Number	Study Title	Tumour	Intervention	Sample Size	Drug
NCT05890781 (recruiting)	Engineering Immune Organoids Study Pediatric Cancer	Sarcoma, brain tumour, kidney tumour, neuroblastoma	Fresh tumour sample to engineer immune organoids from paediatric patient tissues using induced pluripotent stem cells (iPSCs)	100	Not specified
NCT04986748 (recruiting)	Using QPOP to Predict Treatment for Sarcomas and Melanomas	Sarcomas and melanomas	Tumour samples for two- and three-dimensional models to evaluate drug sensitivities ex vivo	100	Panel of 14 drugs
NCT02910895 (recruiting)	A Platform of Patient Derived Xenografts (PDX) and 2D/3D Cell Cultures of Soft Tissue Sarcomas	Soft-tissue sarcoma	Sarcoma-patient-derived xenografts (sarcoma PDXs)	54	Not specified
NCT03358628 (not yet recruiting)	Patient-derived Xenograft (PDX) Modeling to Test Drug Response for High-grade Osteosarcoma	Osteosarcoma	Molecular profiling and in vivo drug testing in PDX	Unknown	
NCT03896958 (recruiting)	The PIONEER Initiative: Precision Insights On N-of-1 Ex Vivo Effectiveness Research Based on Individual Tumour Ownership (Precision Oncology)	Cancer	Functional precision testing of a patient’s tumour tissue to help guide optimal therapy (organoid, drug screening approaches in addition to traditional genomic profiling)	1000	Not specified
NCT05537844 (recruiting)	Longitudinal Sample Collection to Investigate Adaptation and Evolution of Ovarian High-grade Serous Carcinoma	Ovarian cancer (sarcoma, ovarian)	To acquire tumour material at diagnosis and relapse, whole blood for genomic analysis, plasma for ctDNA, and to isolate single cells and establish organoid cultures	250	Not specified
NCT04931381 (recruiting)	Organoid-Guided Chemotherapy for Advanced Pancreatic Cancer	Pancreatic cancer	Organoid test	100	Chemotherapy gemcitabine, 5-fluorouracil, paclitaxel, oxaliplatin, irinotecan
NCT04931394 (recruiting)	Organoid-Guided Adjuvant Chemotherapy for Pancreatic Cancer	Pancreatic cancer	Organoid drug test	200	Adjuvant chemotherapy
NCT05351398 (not yet recruiting)	The Clinical Efficacy of Drug Sensitive Neoadjuvant Chemotherapy Based on Organoid Versus Traditional Neoadjuvant Chemotherapy in Advanced Gastric Cancer	Advanced gastric cancer	Organoid drug test	54	Patient-derived, organoid-based, drug-sensitive, neoadjuvant chemotherapy
NCT05378048 (withdrawn)	Patient-derived-organoid (PDO) Guided Versus Conventional Therapy for Advanced Inoperable Abdominal Tumors	Advanced inoperable abdominal tumours	Organoid test	140	Not specified, genome-guided drug screening
NCT05352165 (not yet recruiting)	The Clinical Efficacy of Drug Sensitive Neoadjuvant Chemotherapy Based on Organoid Versus Traditional Neoadjuvant Chemotherapy in Advanced Rectal Cancer	Rectal cancer	Organoid drug test	190	Standard long-term therapy, wit the addition of FOLFOX; FOLFIRI; or 5-FU; or, 5-FU and pembrolizumab; and other individualised treatments
NCT05024734 (recruiting)	Guiding Instillation in Non Muscle-invasive Bladder Cancer Based on Drug Screens in Patient Derived Organoids	Bladder cancer	Organoid drug test	30	Epirubicin, mitomycin, gemcitabine, docetaxel
NCT05267912 (recruiting)	Prospective Multicenter Study Evaluating Feasibility and Efficacy of Tumor Organoid-based Precision Medicine in Patients With Advanced Refractory Cancers	Advanced, pretreated solid tumours	Organoid drug test	1919	Not specified, panel consisting of chemotherapy, hormonal therapy, targeted therapy
NCT04611035 (recruiting)	Q-GAIN (Using Qpop to Predict Treatment for GAstroIntestinal caNcer)	Gastrointestinal cancer	Organoid drug test	100	Panel of 14 drugs
NCT04450706 (recruiting)	Functional Precision Oncology for Metastatic Breast Cancer	Breast cancer HER2-negative, breast cancer	Organoid drug test	15	Not specified, individualised panels based on genomic analysis and NCCN guidelines
NCT04842006 (recruiting)	Systemic Neoadjuvant and Adjuvant Control by Precision Medicine in Rectal Cancer (SYNCOPE)	Colorectal cancer	Organoid drug test	93	Not specified neoadjuvant therapy and long radiation therapy
NCT05432518 (recruiting)	GBM Personalized Trial (Pilot Trial for Treatment of Recurrent Glioblastoma)	Recurrent glioblastoma	Organoid drug test	10	Afatinib, dasatinib, palbociclib, everolimus, olaparib
NCT05381038 (not yet recruiting)	Optimizing and Personalizing Azacitidine Combination Therapy for Treating Solid Tumors QPOP and CURATE.AI	Gastrointestinal cancer, breast Cancer	Organoids evaluated with QPOP	10	Azacitidine in combination with docetaxel, paclitaxel, or irinotecan
NCT05473923 (recruiting)	PTCs-based Precision Treatment Strategy on Recurrent High-grade Gliomas	Recurrent high-grade glioma	Patient-derived tumour-like cell clusters (multicellular spheroid model)	30	Non-specified, receiving chemotherapeutic or targeted drugs recommended by molecular tumour board

## Data Availability

Not applicable.

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
