# Peer review of "Combination of Genomic Landsscape and 3D Culture Functional Assays Bridges Sarcoma Phenotype to Target and Immunotherapy"

_cells, 2023, doi:10.3390/cells12172204_

Round 1

Reviewer 1 Report

This is a well organized and detailed analysis of how to get more information for response prediction in sarcomas. It presents information on assays using organoids in particular as a means to more rapidly get data compared to N=1 trials.

In terms of immunotherapy I suggest also highlighing challenges of overcoming tumor inhibitory micro environment (TIME) as well as individulized HLA -fit for excellent antigen presentation of cancer -specific sarcoma fusion peptides.

References : Sellars et al (TIME)- Cell 2022 attached

Anderson et al for EWS fusions- Anderson PM, Tu ZJ, Kilpatrick SE, Trucco M, Hanna R, and Chan T. Routine EWS-Fusion analysis in the oncology clinic to identify cancer-specific peptide sequence patterns that span breakpoints in Ewing sarcoma and DSRCT. 2023 Cancers 15:1623 PMID 36900411 https://doi.org/10.3390/cancers 15051623

Author Response

Reviewer 1

This is a well organized and detailed analysis of how to get more information for response prediction in sarcomas. It presents information on assays using organoids in particular as a means to more rapidly get data compared to N=1 trials.

In terms of immunotherapy I suggest also highlighing challenges of overcoming tumor inhibitory micro environment (TIME) as well as individulized HLA -fit for excellent antigen presentation of cancer -specific sarcoma fusion peptides.

References : Sellars et al (TIME)- Cell 2022 attached

Anderson et al for EWS fusions- Anderson PM, Tu ZJ, Kilpatrick SE, Trucco M, Hanna R, and Chan T. Routine EWS-Fusion analysis in the oncology clinic to identify cancer-specific peptide sequence patterns that span breakpoints in Ewing sarcoma and DSRCT. 2023 Cancers 15:1623 PMID 36900411 https://doi.org/10.3390/cancers 15051623

We thank the reviewer for this valuable suggestion and now address this in section 3.3 Current immunotherapy, as follow: The immune tumour microenvironment (TME) and expression of its markers differs by type of sarcoma. Generally, the TME evolves to impede immunity (Sellars). Sarcomas with a high frequency of chromosome copy number alterations, such as UPS and MFS, may be capable of eliciting an immune response and respond better to ICIs or other immunotherapies (Nakata) compared to other sarcoma histotypes because the presence of a high number of genomic alterations that reduce immune suppression at the tumour site. To verify this, Petitprez et al. developed a new classification of STS based on the TME composition in large STS cohorts. Tumours were assigned to one of five sarcoma immune classes (SICs) based on the microenvironment cell population (MCP) determined by the FACS counter. SIC predicted responses to PD-1 blockade in a multicenter Phase II clinical trial of PD-1 blockade therapy (pembrolizumab) in STS (NCT02301039). (Petitprez)

Reviewer 2 Report

Suggestion: major revision. 

This review extensively explores the topics of patient-derived 3D models and precision medicine, particularly focusing on the associated clinical trials. It provides a novel perspective on the synergy between cancer therapy and the genomic landscape.

However, it's crucial to acknowledge that the paper exhibits widespread issues with grammar, formatting, and punctuation throughout. Numerous sections within the article require the appropriate use of colons, and in some instances, colons are inaccurately employed. Additionally, the correct use of italics for terms such as "in vitro" and "ex vivo" needs to be consistent. A comprehensive review of language and vocabulary is essential to elevate the paper's professionalism and readability.

Furthermore, the article underscores the correlation between precision medicine and patient-derived 3D models. However, certain sections of the article lack relevance to precision medicine or 3D models. For instance, the segment on Clinical results of targeted therapy based on the genomic landscape focuses solely on oncology studies and potential targets for cancer therapy. To enhance clarity and coherence, it is recommended that the authors reframe the article's structure or rewrite these sections to ensure a more centered paper aligned with the primary subject matter.

I have other comments below:

Abstract

1. The article discusses various cancer types, yet the title and abstract only mention sarcomas. The authors should revise the title and abstract to accurately reflect the content.

2. The abstract only states that current clinical studies will be reviewed. However, a significant portion of the content does not relate to clinical studies, and this issue should be addressed.

Introduction

3The transition using "Therefore" in line 37 after the preceding sentence lacks logical cohesion.

4Sentences from line 47 to 52 contain significant grammatical mistakes that hinder understanding. The author should revise and correct these sentences.

5. The term "large studies" is not suitable in line 42.

6. While the authors introduce the concept of the organoid model and discuss its advantages compared to clinical trials, it's essential to mention other drug screening platforms and explain why the organoid model should be preferred. This enhancement will facilitate reader understanding of how these models play a pivotal role in assessing drug responses and guiding therapeutic decisions

Individualized patient models for preclinical testing 

7The authors primarily focus on the preclinical testing of various individualized patient models, yet the subtitles mention "in clinical trials." These statements are inaccurate, and a clear separation between preclinical and clinical data is essential to prevent confusion.

8. The authors should place more emphasis on explaining the intricate connection between precision medicine and 3D models. Currently, the discussion primarily revolves around different models and their usage in translational cancer studies. While Table 1 provides a concise summary, it lacks sufficient explanation.

Other potential evolutions 

9. This section does not align with the concept of the genomic landscape or other parts of this review. The authors should consider deleting this entire section.

Discussion and prospective 

10In lines 534 to 542, the authors discuss PDOs (patient-derived organoids) but quickly shift to the broader topic of 3D models. If the authors intend to discuss 3D models, they should omit the initial mention of PDOs or alternatively, focus on PDOs within the limitations section.

11. A refinement of the logical flow is suggested, particularly regarding the transition from discussing challenges to introducing the concept of 3D tumor models (lines 527-531), as well as the transition from the discussion on standardized quality criteria to addressing the complexities of heterogeneity (lines 543-546).

It's crucial to acknowledge that the paper exhibits widespread issues with grammar, formatting, and punctuation throughout. Numerous sections within the article require the appropriate use of colons, and in some instances, colons are inaccurately employed. Additionally, the correct use of italics for terms such as "in vitro" and "ex vivo" needs to be consistent. A comprehensive review of language and vocabulary is essential to elevate the paper's professionalism and readability.

Author Response

Furthermore, the article underscores the correlation between precision medicine and patient-derived 3D models. However, certain sections of the article lack relevance to precision medicine or 3D models. For instance, the segment on Clinical results of targeted therapy based on the genomic landscape focuses solely on oncology studies and potential targets for cancer therapy. To enhance clarity and coherence, it is recommended that the authors reframe the article's structure or rewrite these sections to ensure a more centered paper aligned with the primary subject matter.

Our review focuses on the progress made in precision medicine and discusses various preclinical assays that may improve the choice of treatment for individual patients. Targeted treatment of specific genetic mutations falls under the definition of precision medicine and should benefit from such assays. We have therefore included a brief summary of current therapies. The reviewer is correct in stating that this section focuses solely on oncology targets for cancer therapies. As pointed out below, this and the main focus on sarcomas is due to the fact that the article is under consideration for a special issue on sarcomas. Including non-cancer related genetic defects would have resulted in much longer and less focused paper. For the same reason, we limited the review mainly to recent studies and trials, and on potential future evolutions of preclinical assays. We have therefore not restructured or refocused the paper. We have, however, revised the text in response to specific criticisms.

I have other comments below:

Abstract

  1. The article discusses various cancer types, yet the title and abstract only mention sarcomas. The authors should revise the title and abstract to accurately reflect the content.

The reviewer may not have been made aware of the fact that the article was submitted for a special issue of Cells on sarcomas. Our review does occasionally look over the fence to other tumours for two reasons. First, because the challenges to establish the treatment most likely to benefit an individual patient are the same for other rare and genetically diverse cancers, for which no large clinical trial are possible.  Second, because very limited data have been reported on some of the assay types discussed here, or because trials including them are still in progress. Nevertheless, the main focus of the review is on sarcomas. The title is therefore appropriate, and the fact that the paper will also be of interest for other rare cancers is mentioned in the abstract. 

  1. The abstract only states that current clinical studies will be reviewed. However, a significant portion of the content does not relate to clinical studies, and this issue should be addressed.

The revised abstract now reads: Here, we provide an overview of current clinical studies that combine genomic targeted therapy with patient-derived 3D models as drug assay platform. This approach opens new avenues to predict responses to individualized treatment when genomic and pathway alterations are not sufficient to guide the choice of the most promising treatment.

Introduction

  1. The transition using "Therefore" in line 37 after the preceding sentence lacks logical cohesion.

We had noted this and now begin the sentence with “Furthermore”

  1. Sentences from line 47 to 52 contain significant grammatical mistakes that hinder understanding. The author should revise and correct these sentences.

 We corrected these and many other linguistic errors, but have not highlighted these edits

  1. The term "large studies" is not suitable in line 42.

We deleted “large” Corrected in ms

  1. While the authors introduce the concept of the organoid model and discuss its advantages compared to clinical trials, it's essential to mention other drug screening platforms and explain why the organoid model should be preferred. This enhancement will facilitate reader understanding of how these models play a pivotal role in assessing drug responses and guiding therapeutic decisions.

As suggested we now include the following statement: To identify suitable personalized drug treatments, 3D models are preferable to traditional 2D-monolayer cell cultures, because they better reflect the tumour heterogeneity, the interactions between tumour cells and extracellular matrix, and the tumour microenvironment. Similarly, PDX models in which fresh patient tumour tissues are directly transplanted into immunocompromised mice are superior to standard cell line-derived xenografts because they maintain the histological, epigenetic and genetic characteristics.

Corrected the modified new text in ms. Also had to modify the following sentence, because the logical connection was disrupted by the sentence on PDX. It now reads: This requires the development of new assays, in particularIndividualized patient models for preclinical testing 

  1. The authors primarily focus on the preclinical testing of various individualized patient models, yet the subtitles mention "in clinical trials." These statements are inaccurate, and a clear separation between preclinical and clinical data is essential to prevent confusion.

We correct the title of chapter  5.2 to read: Spheroid assays to test chemo- and targeted therapy in preclinical and clinical trials

  1. The authors should place more emphasis on explaining the intricate connection between precision medicine and 3D models. Currently, the discussion primarily revolves around different models and their usage in translational cancer studies. While Table 1 provides a concise summary, it lacks sufficient explanation.

The connection was already amply addressed by Figure 1 and the accompanying text of the Introduction. Table 1 only serves to provide an overview of the studies of section 2, which follows the Introduction

  1. Other potential evolutions. This section does not align with the concept of the genomic landscape or other parts of this review. The authors should consider deleting this entire section.

We  corrected in manuscript

It does align, in so far as it provides another way by which 3D models could improve cancer treatment and drug screening. As far as we know, this has not been previously proposed, so we would prefer not to delete it.

Discussion and prospective 

  1. In lines 534 to 542, the authors discuss PDOs (patient-derived organoids) but quickly shift to the broader topic of 3D models. If the authors intend to discuss 3D models, they should omit the initial mention of PDOs or alternatively, focus on PDOs within the limitations section.

The sentence now reads: The development and implementation of PDOs and other assays to evaluate drug vulnerability of individual patients and guide their treatment have not yet received clinical validation.  

  1. A refinement of the logical flow is suggested, particularly regarding the transition from discussing challenges to introducing the concept of 3D tumor models (lines 527-531), , as well as the transition from the discussion on standardized quality criteria to addressing the complexities of heterogeneity (lines 543-546)

The chapter essentially lists a number of necessary steps and does to attempt to have smooth transitions between them. To make this less obvious, we changed the title of section 7 from “Discussion and Prospective” to “Prospective”. Corrected in manuscript

Comments on the Quality of English Language: It's crucial to acknowledge that the paper exhibits widespread issues with grammar, formatting, and punctuation throughout. Numerous sections within the article require the appropriate use of colons, and in some instances, colons are inaccurately employed. Additionally, the correct use of italics for terms such as "in vitro" and "ex vivo" needs to be consistent. A comprehensive review of language and vocabulary is essential to elevate the paper's professionalism and readability.

We apologize for the many errors in the previous manuscript. The last author, Prof. Palinski, who has worked at the university of California  since 1986, has now extensively edited the text. All in vitro”  and ”in vivo”  are now in italcs

Reviewer 3 Report

The authors of this manuscript provide a comprehensive review about the role of genomic profiling and 3D models in targeting sarcomas with most appropriate therapies. The topic is of great interest and the manuscript is well organised, however some topics should be discussed in more details to add further value to the manuscript.

1) Genomic profiling is a powerful and effective tool to identify mutations and fusion transcripts which can be targeted by specific drugs, as mentioned by the authors. However, another interesting aspect of genomic profiling - particularly in combination with pharmacological testing of patient-derived cell models- is that of identifying chemoresistant signatures which would allow treatment modulation driven by a biomarker-based patient stratification. In this regard a recent translational study aimed at investigating intra- and inter-genomic variability of MFS and UPS by combining patient-derived primary cultures, pharmacological profiling and RNAseq identified the involvement of several immunoglobulin genes, neutrophil-mediated immunity and activation pathways in chemoresistant cell lines. Please briefly discuss this aspect including relevant supporting references: doi: 10.1186/s12967-015-0466-4, doi.org/10.3390/ijms24086926, doi: 10.3892/or.2021.8086.

2) in vitro 3D tumor models are highly helpful in recapitulating some of the most relevant features of sarcomas, making them a great tool for translational studies investigating chemosensitivity and drug screening. Besides organoids and spheroids however, also some other 3D models could be relevant for this goal.

For example, collagen-based scaffoldare widely used as excellent ECM mimetic devices. Indeed, recent studies demonstrated the capability of collagen-based scaffolds to induce different drug resistance mechanisms and pathological features on the basis of cell types (doi: 10.1016/j.ceb.2010.08.015,  doi: 10.1177/17588359221093973doi: 10.3390/ijms222111564). 

Moreover, increasing interest is related to finding novel high-throughput strategies combining microfluidics, bioengineering, nanotechnology and the use of cells or tissue for the development of lab-on-a-chip devices as promising platforms for drug screening (doi: 10.3389/fbioe.2022.953555, doi:10.1039/d0lc00424c, doi:10.1038/s41598-019-38666-9).

Overall English is ok, however there are several grammar mistakes and typos throughout the text as well as some sentences without full meaning. Please revise carefully the text. 

Author Response

Reviewer 3

The authors of this manuscript provide a comprehensive review about the role of genomic profiling and 3D models in targeting sarcomas with most appropriate therapies. The topic is of great interest and the manuscript is well organised, however some topics should be discussed in more details to add further value to the manuscript.

We thank the reviewer for the helpful comments.

1) Genomic profiling is a powerful and effective tool to identify mutations and fusion transcripts which can be targeted by specific drugs, as mentioned by the authors. However, another interesting aspect of genomic profiling - particularly in combination with pharmacological testing of patient-derived cell models- is that of identifying chemoresistant signatures which would allow treatment modulation driven by a biomarker-based patient stratification. In this regard a recent translational study aimed at investigating intra- and inter-genomic variability of MFS and UPS by combining patient-derived primary cultures, pharmacological profiling and RNAseq identified the involvement of several immunoglobulin genes, neutrophil-mediated immunity and activation pathways in chemoresistant cell lines Please briefly discuss this aspect including relevant supporting references: doi: 10.1186/s12967-015-0466-4, doi.org/10.3390/ijms24086926, doi: 10.3892/or.2021.8086.

We agree and now discuss this in section 3 (Clinical results of target therapy based on genomic landscape), as follows: Targeted therapy is primarily directed against genomic rearrangements. Another interesting aspect of genomic profiling is the identification of chemoresistant signatures which would allow treatment modulation driven by a biomarker-based patient stratification and patient-derived cell culture assays. Recent studies investigating intra- and inter-genomic variability of undifferentiated pòlymorphic sarcoma and myxoid fibrosarcoma identified the involvement of several immunoglobulin genes, neutrophil-mediated immunity and activation pathways in chemoresistant cell lines (Vanni, Foley, Iwata). They also investigated intra- and inter-genomic variability of these two sarcoma histotypes, which could partially determine susceptibility or resistance to chemotherapies.

2) in vitro 3D tumor models are highly helpful in recapitulating some of the most relevant features of sarcomas, making them a great tool for translational studies investigating chemosensitivity and drug screening. Besides organoids and spheroids however, also some other 3D models could be relevant for this goal.

For example, collagen-based scaffolds are widely used as excellent ECM mimetic devices. Indeed, recent studies demonstrated the capability of collagen-based scaffolds to induce different drug resistance mechanisms and pathological features on the basis of cell types (doi: 10.1016/j.ceb.2010.08.015,  doi: 10.1177/17588359221093973, doi: 10.3390/ijms222111564). 

We now discuss this in section 6 (Conclusion and Prospective), as follows: Synthetic polymers, such as hydrogels or other materials with well-defined structure and inert properties may overcome alteration in growth [115]. Collagen-based scaffolds better mimicking the microenvironment of some rare sarcomas are another option (Vanni 2022. Egeblad 2010).

3) Moreover, increasing interest is related to finding novel high-throughput strategies combining microfluidics, bioengineering, nanotechnology and the use of cells or tissue for the development of lab-on-a-chip devices as promising platforms for drug screening (doi: 10.3389/fbioe.2022.953555, doi:10.1039/d0lc00424c, doi:10.1038/s41598-019-38666-9).

We discuss this point in section 5.6 (Explants on chip sensor (Organ on Chip) assays in clinical trials) as follows: Several novel approaches are currently being explored that enable both automation and high-throughput strategies combining microfluidics, bioengineering, and  nanotechnology for the development of lab-on-a-chip devices, and may lead to promising platforms for drug screening (Mercatelli 2022; Chramiec 2020)

4) There are several grammar mistakes and typos throughout the text as well as some sentences without full meaning. Please revise carefully the text. 

We have carefully revised the text

Round 2

Reviewer 2 Report

The comments have been effectively handled, and the manuscript appears well-present. 

Author Response

 Thanks for reviewer's comment

I corrected all points indicated
